# Composition and Antioxidant Activity of Anthocyanins and Non-Anthocyanin Flavonoids in Blackberry from Different Growth Stages

**DOI:** 10.3390/foods11182902

**Published:** 2022-09-18

**Authors:** Jing Li, Chong Shi, Dongbei Shen, Tianyu Han, Wenlong Wu, Lianfei Lyu, Weilin Li

**Affiliations:** 1Co-Innovation Center for Sustainable Forestry in Southern China, College of Forestry, Nanjing Forestry University, Nanjing 210037, China; 2Jiangsu Key Laboratory for the Research and Utilization of Plant Resources, Institute of Botany, Jiangsu Province and Chinese Academy of Sciences, Nanjing 210014, China; 3College of Light Industry and Food Engineering, Nanjing Forestry University, Nanjing 210037, China

**Keywords:** blackberry, flavonoids, anthocyanin, different ripening stages

## Abstract

The high nutritional value and unique flavor of blackberries make them a popular food choice among consumers. Anthocyanin content (AC) and non-anthocyanin flavonoid content (NAFC) are important functional components in blackberry. We tested the AC, NAFC, and antioxidant activities of two blackberry—Ningzhi 1 and Hull—during the following ripening stages: green-fruit stage (GFS), color-turning stage (CTS), reddening stage (RDS), and ripening stage (RPS). The results showed that NAFC decreased and AC increased gradually during the ripening stages. The NAFC of Hull blackberry was the highest during GFS (889.74 μg/g), while the AC of Ningzhi 1 blackberry was the highest during RPS (1027.08 μg/g). NAFC was the highest at the initial stage and gradually decreased with ripening. Anthocyanin accumulation mainly occurred during the later ripening stages. These results provide a reference for comparing the NAFC, AC, and antioxidant activity of Ningzhi 1 and Hull and their changes during different ripening stages.

## 1. Introduction

Blackberry (*Rubus* spp.), belonging to the genus *Rubus* of the Rosaceae family, is a perennial shrub with aggregate fruit. Blackberries were introduced to China from the United States and mainly grow in Jiangsu, Guizhou, and Sichuan provinces in China [1]. As blackberry fruit cannot be stored for more than 2 days under normal storage conditions, most blackberry fruit is used to make frozen fruit, jam, juice, fruit wine, and other products [2,3]. Therefore, the quality of blackberries is very important to consumers as well as the food-processing industry. Both the external and internal qualities of the fruit directly influence the consumer’s desire for fruit consumption, and high-quality fruit has a higher market value [4].

In recent years, consumers have shown increasing interest in blackberries due to its rich nutrient content. Blackberry, as one of the third-generation new special fruit recognized by the United Nations Food and Agriculture Organization, is a rich source of minerals, vitamins (such as vitamins E and C), calcium, flavonoids, anthocyanins and ellagic acid [5]. Modern pharmacological studies showed that eating fruit with high anthocyanin content (AC) and other phenolic substances can not only alleviate aging but also reduce the risk of chronic diseases [6,7]. The high antioxidant properties of blackberries were shown to correlate with their anthocyanins and flavonoid contents [8]. Researchers have also demonstrated that AC in ripe blackberry was significantly higher than that in red raspberry [9]. Blackberry fruit contains approximately 15 phenolic substances, particularly anthocyanin-3-*O*-glucoside and chlorogenic acid [10]. Previous research conducted a phytochemical evaluation of different blackberry cultivars (Jumbo, Black Satin, and Dirksen) using HPLC-TOF-MS and identified anthocyanins cyanidin-3*-O-*glucoside, cyanidin-3*-O-*xyloside, cyanidin-3*-O-*(6*-O-*malonyl glucoside), cyanidin-3*-O-*β-(600-(3-hydroxy-3-methylglutaroyl)-glucoside, flavonols (quercetin-3*-O-*rutinoside), hydrolyzable tannins (ellagitannins, lambertianin C, and dimer of galloyl-diHHDP-glucose), and condensed tannins with different degrees of polymerization [11].

A research on blackberry cv Boysen and Arapaho showed that the aggregate fruit formed at 9 days after flowering (DAF) and there were four ripening stages: green fruit stage (green berries), color-turning stage (green–pink berries), reddening stage (red berries), and ripening stage (purple berries). The color of the blackberries changed from green to mottled red at approximately 20 DAF and changed fully red at approximately 30 DAF. At 40 DAF, the blackberries fully ripened and the color changed to purple [12]. Different blackberry cultivars mature at different times, from May to July, and harvest duration varies from 4 to 14 days. Blackberries for the fresh fruit market are primarily harvested by hand to maintain postharvest quality, although there has been some research on tendon-driven soft robotic gripper for the harvesting of blackberries [13]. The blackberry picking period influences its quality, as the correct picking period can result in high-quality blackberry fruit. However, at present, there lacks systematic investigation on the changes in the content of antioxidants, flavonoids and anthocyanins during the fruit ripening stages, as well as on the monitoring of differences in the growth stages of different cultivars.

In this study, two blackberry cultivars, Ningzhi 1 and Hull, were selected to evaluate changes in anthocyanin content (AC), non-anthocyanin flavonoid content (NAFC), and antioxidant capacity of the fruit during the green fruit stage (GFS), color-turning stage (CTS), reddening stage (RDS) and ripening stage (RPS). In addition, the relationship between these non-anthocyanin flavonoids and anthocyanins and changes in antioxidant indices provides a theoretical basis for blackberry fruit harvest.

## 2. Materials and Methods

### 2.1. Preparation of Samples

As shown in Figure 1, two blackberry cultivars, Ningzhi 1 and Hull, were cultivated at experimental sites at the Institute of Botany, Jiangsu Province and Chinese Academy of Sciences in Lishui, Nanjing, Jiangsu Province, China (119°07′ E, 31°28′ N). The day/night temperature in the greenhouse was maintained at 28/19 °C with 70% relative humidity. Blackberry samples were harvested at four developmental stages: 10, 20, 30 and 40 DAF. The four ripening stages of Ningzhi 1 blackberry were named green fruit stage Ningzhi 1 (GFSN), color-turning stage Ningzhi 1 (CTSN), reddening stage Ningzhi 1 (RDSN) and ripening stage Ningzhi 1 (RPSN), while those of Hull blackberry were named green fruit stage Hull (GFSH), color-turning stage Hull (CTSH), reddening stage Hull (RDSH), and ripening stage Hull (RPSN). The selected fruit was treated by liquid nitrogen and then stored at −80 °C for experiments. At least 24 specimens per sample were used and the experiment was repeated three times.

### 2.2. Separation of Anthocyanins and Non-Anthocyanin Flavonoid Fractions

Anthocyanin and non-anthocyanin flavonoid fractions of blackberry were separated according to a previous study [14]. Samples (2 g) were ground in a grinder for 2 min, extracted with 20 mL of acetone solution for 2 h, and centrifuged (8000× *g*, 15 min) to collect a supernatant. Then, 15 mL of supernatant was mixed with 10 mL of the ligarine vortex for 30 s to remove lipids. After the degreasing process was completed, the supernatant (10 mL) was mixed with 0.5 mL HCl. Finally, the supernatant (10 mL) was added into the C18 SepPak^®^ (400 mg packing, Waters, Milford, MA, USA) cartridge for 30 min of adsorption, followed by desorption with 10 mL deionized water to obtain the blackberry anthocyanin fractions, and finally blackberry flavonoid fractions with 10 mL ethyl acetate were obtained.

### 2.3. UPLC-MS/MS Analysis

The anthocyanins and non-anthocyanin flavonoid fractions were analyzed using the ExionLC™ AD system (SCIEX, Framingham, MA, USA) coupled with a QTRAP^®^6500+ mass spectrometer (SCIEX, Framingham, MA, USA). The chromatographic separation was performed using an ACQUITY BEH C18 (100 mm × 2.1 mm, 1.7 µm, Waters, Milford, MA, USA), separation column temperature 40 °C, flow rate 0.35 mL/min, and injection volume 2 μL. The mobile phase was (A) 0.1% formic acid (*v/v*) and (B) 0.1% formic acid ethanol solution (*v/v*) with gradient elution of 5% B (0 min), 50% B (6.5 min), 95% B (12.5 min), and 5% B (14 min). The conditions of MS/MS were electrospray positive ion mode: voltage 5500 V, temperature 550 °C, and curtain gas 35 psi.

### 2.4. Free-Radical-Scavenging of 2,2’-Azino-bis(3-ethylbenzothiazoline-6-sulfonic Acid) (ABTS)

The ABTS radical-scavenging activities of blackberry anthocyanin and non-anthocyanin flavonoid fractions were determined using a previously described method [15]. ABTS+ solution preparation: 5 mL of ABTS solution (7 mM) reacted with 5 mL of K_2_S_2_O_8_ (2.45 mM) and incubated in the dark at room temperature for 12 h until the reaction was complete. The solution of ABTS+ was then diluted with 80% ethanol (*v/v*) to OD_734_ of 0.700 ± 0.020. Thereafter, the sample (20 μL) and ABTS+ solution (1.5 mL) were mixed. After reaction at room temperature for 6 min, OD_734_ was measured. The ABTS free-radical scavenging rate of blackberry anthocyanins and non-anthocyanin flavonoid fractions was calculated using Equation (1):(1)$$\mathrm{Free}-\mathrm{radical}\;\mathrm{scavenging}\;(\%)=\frac{A_{0}-(A_{j}-A_{i})}{A_{0}}\times 100\%$$
where A_0_ is absorbance of control, A_j_ is absorbance of test sample, and A_i_ is absorbance of sample.

### 2.5. Free Radical Scavenging of 2,2-Diphenyl-1-picrylhydrazyl (DPPH)

DPPH radical-scavenging activities of blackberry anthocyanin and non-anthocyanin flavonoid fractions were determined using a previously reported method [16]. The sample (20 µL) was mixed with 150 µM DPPH solution (200 µL). After 30 min reaction at room temperature, OD_510_ was measured. The DPPH free-radical scavenging rate of blackberry anthocyanin and non-anthocyanin flavonoid fractions were calculated using Equation (1).

### 2.6. Free-Radical Scavenging of Hydroxyl

The hydroxyl radical-scavenging activities of blackberry anthocyanin and non-anthocyanin flavonoid fractions were determined using the Fenton method [17]. Briefly, the sample (1 mL), 2 mL of FeSO_4_ (4 mM) and 1 mL of salicylic acid–ethanol solution (7 mM) were mixed, 0.5 mL H_2_O_2_ solution (4 mM) added, and incubated in a water bath at 37 °C for 30 min. After the completion of the reaction, OD_510_ was measured. The hydroxyl radical-scavenging rates of the blackberry anthocyanin fractions and blackberry flavonoid fraction were calculated using Equation (1).

### 2.7. Free-Radical Scavenging of Superoxide Anions

The superoxide anion radical-scavenging activities of blackberry anthocyanins and non-anthocyanin flavonoid fractions were determined using the pyrogallol method [18]. Briefly, 3 mL Tris-HCl (pH = 8.2, 0.05 M) solution was placed in a water bath at 25 °C for 20 min, 1 mL of sample was added, and then 0.5 mL pyrogallol (0.025 M) was added. After 6 min of reaction, OD_320_ was measured. The superoxide anion radical-scavenging rates of blackberry anthocyanin and flavonoid fractions were calculated using Equation (1).

### 2.8. Reagents

Acetone (CAS number: 67-64-1, AR), ethyl acetate (CAS number: 141-78-6, AR), formic acid (CAS number: 64-18-6, GR), ethanol (CAS number: 64-17-5, GR), ABTS (CAS number: 30931-67-0, AR), DPPH (CAS number: 84077-81-6, AR), salicylic acid (CAS number: 69-72-7, AR), FeSO_4_ (CAS number: 7720-78-7, AR), and HCl (CAS number: 7647-01-0, AR).

### 2.9. Statistical Analysis

All assays were repeated three times. The results are shown as means ± standard deviation. Data were processed using SPSS 23.0 (Chicago, IL, USA). For each cultivar, one-way analysis of variance (ANOVA) and Duncan’s test (*p* < 0.05) were used to determine the significance of differences. All graphs were plotted using Origin 2020 software (Microcal™ Software Inc., Northampton, MA, USA).

## 3. Results and Discussion

### 3.1. Changes in NAFC during Different Ripening Stages

The changes in NAFC during the different ripening stages of Ningzhi 1 and Hull are shown in Table 1. Polyphenols were present as kaempferol-3*-O-*rutinoside, naringenin, naringenin-7*-O-*glucoside, quercetin-3*-O-*glucoside, rutin, procyanidin B2, B3, and C1. There was no difference between Ningzhi 1 and Hull in the types of polyphenols, but there was a significant difference in their content. Some previous studies have demonstrated large differences in the accumulation of bioactive substances in the fruit of different cultivars of blackberry. For example, the total phenolic and flavonoid contents in the ripe fruit of Triple Crown were significantly higher than those of Shawnee [19,20]. In our study, Ningzhi 1 and Hull had the highest NAFC of 492.38 μg/g and 899.74 μg/g, respectively, during the GFS. As the fruit ripened, NAFC gradually decreased. This may be due to the conversion of the polyphenol to anthocyanin fractions and the decreased content of other polyphenol components during the growth stages [21]. The NAFC in Hull was significantly higher than Ningzhi 1 during the ripening stage. In Ningzhi 1, the NAFC, particularly of kaempferol-3*-O-*rutinoside, naringenin, naringenin-7*-O-*glucoside, and quercetin-3*-O-*glucoside, significantly increased from GFS to CTS. In a previous study, the NAFC in the highbush blueberry was shown to be highest during the GFS, and then gradually decreased as the fruit ripened [22]. In contrast, the NAFC in Hull significantly decreased from GFS to CTS, which was different from Ningzhi 1. This may be because the accumulation time of non-anthocyanin flavonoid fractions in Hull is shorter than 20 days, which was not detected in this experiment. In addition, during the fruit-growth stage of blackberry, NAFC in both Ningzhi 1 and Hull fruit gradually decreased. There could be two reasons for this phenomenon. First, the early stage of blackberry development is the main period for the synthesis of phenolic substances [23]. Second, fruit enlargement during blackberry fruit development leads to the dilution of phenolic compounds [24]. The NAFC in Hull was significantly higher than that in Ningzhi 1 during the whole fruit-ripening stage. During fruit ripening, the levels of procyanidins, such as procyanidin B2, B3, and C, in blackberry fruit decreased rapidly. This result is consistent with that of grapes, which may be due to the accumulation of blackberries immediately after fruit setting, with a maximum accumulation level near the discoloration of the blackberries peels, followed by a rapid decline [25]. In general, blackberries had significantly lower levels of NAFC as they ripened, and higher levels of NAFC had to be picked during the early ripening stage. A previous study has reported similar results of a significant increase in NAFC during blackberry ripening [26].

### 3.2. Changes in AC during Different Ripening Stages

Anthocyanins are natural water-soluble pigments of flavonoid derivatives that impart color to flowers and fruit. As shown in Table 2, the anthocyanin fractions detected in blackberry fruit included cyanidin-3*-O-*glucoside, cyanidin-3*-O-*sophoroside, pelargonidin-3*-O-*glucoside, petunidin-3*-O-*glucoside, peonidin-3*-O-*galactoside and peonidin-3*-O-*glucoside. The UPLC-MS/MS results were comparable to those of previous studies [27]. From CTS to RDS of Ningzhi 1 blackberry, the contents of cyanidin-3*-O-*glucoside and cyanidin-3*-O-*sophoroside gradually increased to 332.00 and 402.33, respectively, which was significantly higher than other anthocyanin fractions. From GFS to RPS of Hull, the content of cyanidin-3*-O-*glucoside increased rapidly from 5.52 to 507.67, which was significantly higher than that of the other anthocyanin fractions. Previous studies also confirmed that the content of cyanidin-3*-O-*glucoside increased significantly in the process of blackberry from green fruit to ripening, and was the main component of anthocyanin in blackberry fruit [28]. With the growth of fruit, the content of anthocyanin fractions increased significantly. Similar results have reported that there is a significant increase in total anthocyanin content during blackberry fruit ripening [26]. Furthermore, the comparison the AC of Ningzhi 1 and Hull fruit at different ripening stages showed that the AC differed between the two cultivars. For example, the anthocyanin fractions in Ningzhi 1 fruit mainly consisted of cyanidin-3*-O-*glucoside, cyanidin-3*-O-*sophoroside, and cyanidin-3*-O-*rutinoside, and those of Hullcyanidin-3*-O-*glucoside and cyanidin-3*-O-*xyloside. The differences observed in the anthocyanin fractions of Marion and Evergreen blackberry in different ripening stages exhibited different main AC [29].

### 3.3. Changes in Antioxidant Activity of Non-Anthocyanin Flavonoid Fractions during Different Ripening Stages

Fruit contains a variety of polyphenols, anthocyanin fractions and other antioxidant active substances [30]. To evaluate the antioxidant activity of blackberry fruit at different ripening stages, the free-radical scavenging rates of ABTS, DPPH, hydroxyl and superoxide anion of the non-anthocyanin flavonoid fractions in Ningzhi 1 and Hull were determined. As shown in Figure 2, the antioxidant activities of the non-anthocyanin flavonoid fractions in the fruit of the two cultivars gradually decreased with ripening. During GFS, the non-anthocyanin flavonoid fractions in Ningzhi 1 and Hull had the strongest antioxidant activity. The ABTS, DPPH, hydroxyl, and superoxide anion free-radical scavenging rates of the non-anthocyanin flavonoid fractions in Ningzhi 1 were 48.80%, 49.33%, 62.97%, and 62.66%, respectively, and those in Hull were 83.44%, 91.75%, 92.43%, and 82.43%, respectively. Therefore, the ABTS, DPPH, hydroxyl and superoxide anion free-radical scavenging rates in Hull were higher than those in Ningzhi 1 during the fruit-ripening stage. Studies have demonstrated a strong correlation between antioxidant activity and the non-anthocyanin flavonoid fractions in blackberries [8]. As shown in Table 1, NAFC was significantly reduced during the ripening stages of Ningzhi 1 and Hull. Therefore, the antioxidant capacity of non-anthocyanin flavonoid fractions in Ningzhi 1 and Hull gradually decreased. The highbush blueberry had the highest NAFC in GFS and the lowest content in RPS, and thus the antioxidant activity was consistent with the experimental results [22].

### 3.4. Changes in Antioxidant Activity of Anthocyanin during Different Ripening Stages

Anthocyanins are not only used as food coloring agents but also consumed to prevent cancer, coronary heart disease, and other degenerative diseases [31,32]. ABTS, DPPH, hydroxyl and superoxide anion-scavenging rates were measured to evaluate the antioxidant activity of anthocyanin fractions in Ningzhi 1 and Hull fruit. The comparative analysis results of the antioxidant properties of the anthocyanin fractions at different fruit growth stages are shown in Figure 3. The antioxidant activity of anthocyanin in Ningzhi 1 and Hull fruit increased significantly during the ripening stages. During GFS, the anthocyanin in Ningzhi 1 and Hull had the strongest antioxidant activity. The ABTS, DPPH, hydroxyl, and superoxide anion free-radical scavenging rates of the anthocyanin in Ningzhi 1 were 76.44%, 83.82%, 87.63%, and 80.72%, respectively, and those in Hull were 63.02%, 54.09%, 65.42%, and 61.86%, respectively. During the GFS and CTS, the antioxidant capacity of Ningzhi 1 was weaker than that of Hull, as the AC of Ningzhi 1 was lower. However, during the RDS and RPS, the antioxidant capacity of Ningzhi 1 was stronger than that of Hull. From 20 to 40 days, the anthocyanin fractions in Ningzhi 1 fruit increased rapidly; thus, the antioxidant activity was stronger than that of Hull. Some researchers have reported that the anthocyanin fractions in blackberry fruit are powerful antioxidants [33,34]. As Table 2 exhibits, AC significantly increased during GFS, CTS, RDS, and RPS in Ningzhi 1 and Hull. Therefore, the antioxidant capacity of the anthocyanin fractions in Ningzhi 1 and Hull gradually increased during the ripening stages. In the early ripening stage of Ningzhi 1 and Hull, AC was low, and some polyphenols were converted into anthocyanin fractions in the later stage, thus increasing AC and the antioxidant activity [21]. The antioxidant activity of blackberries at different ripening stages was strongly correlated with NAFC, AC, and the antioxidant activities of blackberry differed in different cultivars.

### 3.5. Principal Component Analysis

Principal component analysis (PCA) of NAFC, AC, non-anthocyanin flavonoid antioxidant activity, and anthocyanin antioxidant activity in Ningzhi 1 and Hull fruit at different ripening stages was conducted. In Figure 4, the PCA results show that PCA obtains two factors (PC−1 and PC−2), which can explain 97.09% of the total variance. PC−1 and PC−2 can explain 48.71% and 48.38% of the variance of the data, respectively. PC−1 with AC (0.829), AA (0.918), AS (0.876), AD (0.917) and ADP (0.860) showed a strong positive correlation. Therefore, RDSN, RPSN, GFSH, CTSH, RDSH, and RPSH were assigned to the left (Figure 4B). GFSN and CTSN were assigned to the right. This outcome may be due to the fact that RDSN, RPSN, GFSH, CTSH, RDSH, and RPSH have higher AC than GFSN and CTSN, as shown in Table 2. In all samples, the groups with close distances have stronger similarity. As shown in Figure 4A, Ningzhi 1 and Hull fruit were divided into four groups based on the ripening stages. This is because the AC of the adjacent Ningzhi 1 and Hull fruit was similar during the growth stages. PC−2 had a strong positive correlation with NAF (0.880), NAFA (0.885), NAFDP (0.907), NAFD (0.888) and NAFS (0.822). GFSN and CTSN were assigned as previously described. This was due to the NAFC in different ripening stages of Ningzhi 1 and Hull. As Table 1 depicts, NAFC significantly reduced during different ripening stages of Ningzhi 1 and Hull. In summary, N40 had the highest AC, whereas H10 can get a higher NAFC. This provides guidance on the best time to harvest blackberry fruit.

## 4. Conclusions

We found that the NAFC was highest during the initial ripening stage, and gradually decreased as the fruit ripened. An accumulation of the anthocyanin fractions occurred mainly in the later ripening stages. There was no difference in the non-anthocyanin flavonoid and anthocyanin fractions of the two cultivars, but there was difference in their content. The antioxidant capacity of the non-anthocyanin flavonoid fractions gradually decreased, while that of anthocyanin fractions gradually increased with fruit ripening. Hull blackberry had the highest NAFC (889.74 μg/g) during the GFS, and the AC of Ningzhi 1 blackberry was the highest during the RPS (1027.08 μg/g). The antioxidant capacity of non-anthocyanin flavonoids plays a key role in the antioxidant capacity of blackberry. Our investigation provides a reference for comparing the changes in the NAFC, AC, and antioxidant capacity of Ningzhi 1 and Hull. This study provides guidance for selecting the best harvest period for Ningzhi 1 and Hull, which will be beneficial in improving their quality for raw consumption, processing, and medicinal purposes.

## Figures and Tables

**Figure 1 foods-11-02902-f001:**
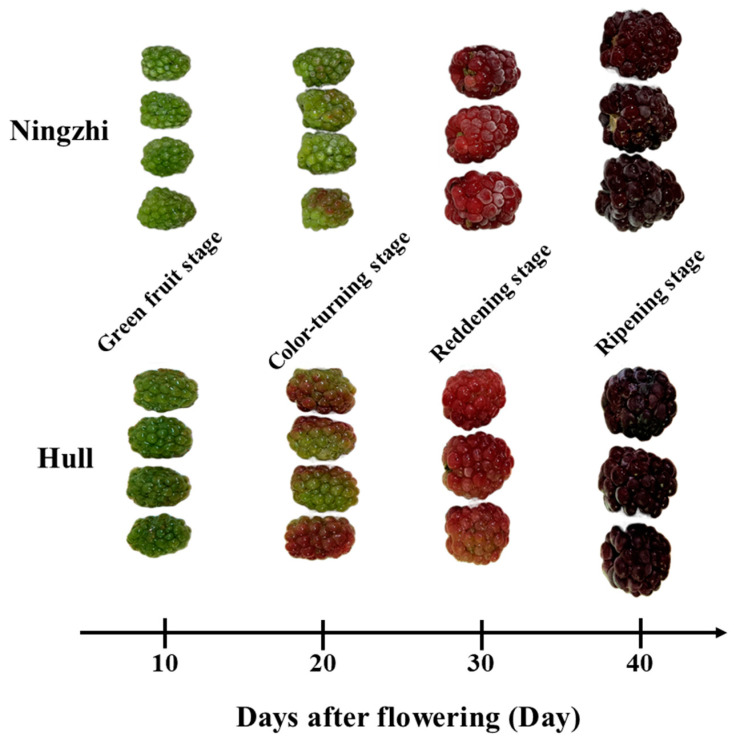
Four growth stages of Ningzhi 1 and Hull.

**Figure 2 foods-11-02902-f002:**
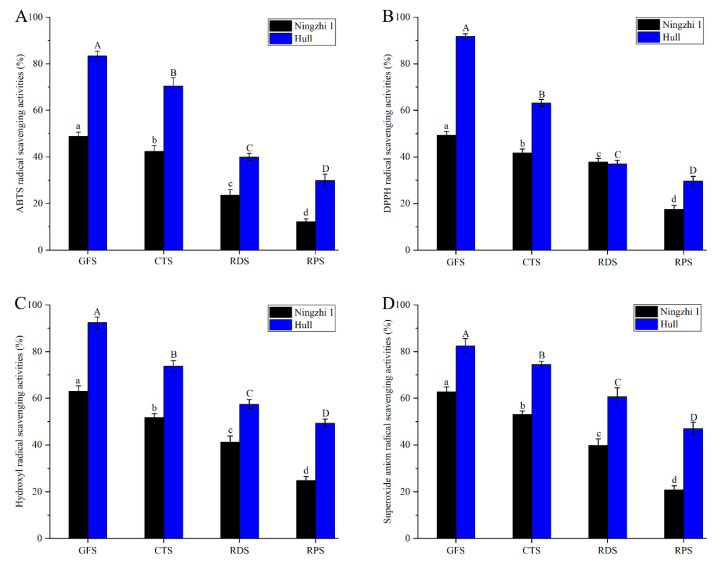
Antioxidant activity of non-anthocyanidin flavonoids in blackberry fruit at different growth stages. (**A**) ABTS radical-scavenging activities. (**B**) DPPH radical-scavenging activities. (**C**) Hydroxyl radical-scavenging activities. (**D**) Superoxide anion scavenging activities. Lowercase letters and uppercase letters mean the significance (*p* < 0.05) of antioxidant activity of non-anthocyanidin flavonoids in Ningzhi 1 and Hull blackberry at different growth stages.

**Figure 3 foods-11-02902-f003:**
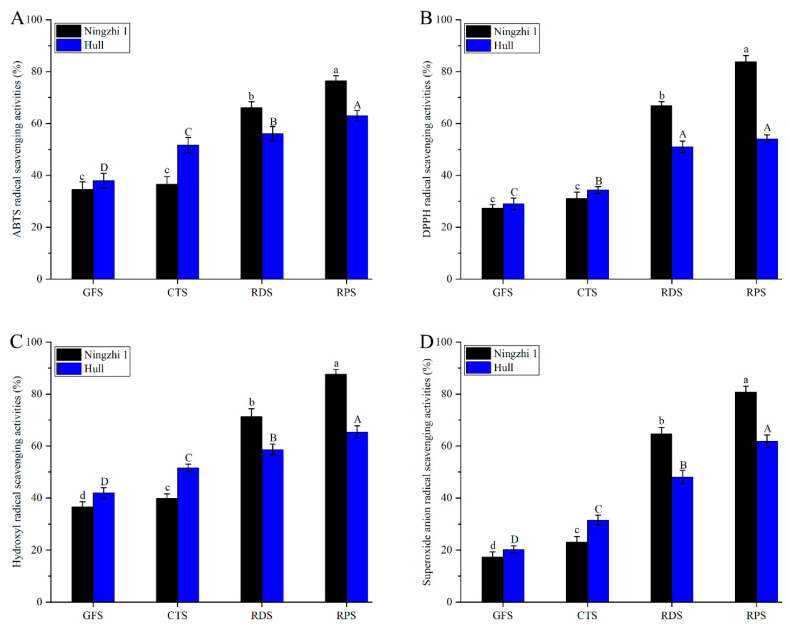
Antioxidant activity of anthocyanidin in blackberry fruit at different growth stages. (**A**) ABTS radical-scavenging activities. (**B**) DPPH radical-scavenging activities. (**C**) Hydroxyl radical-scavenging activities. (**D**) Superoxide anion-scavenging activities. Lowercase letters and uppercase letters mean the significance (*p* < 0.05) of antioxidant activity of non-anthocyanidin flavonoids in Ningzhi 1 and Hull blackberry at different growth stages.

**Figure 4 foods-11-02902-f004:**
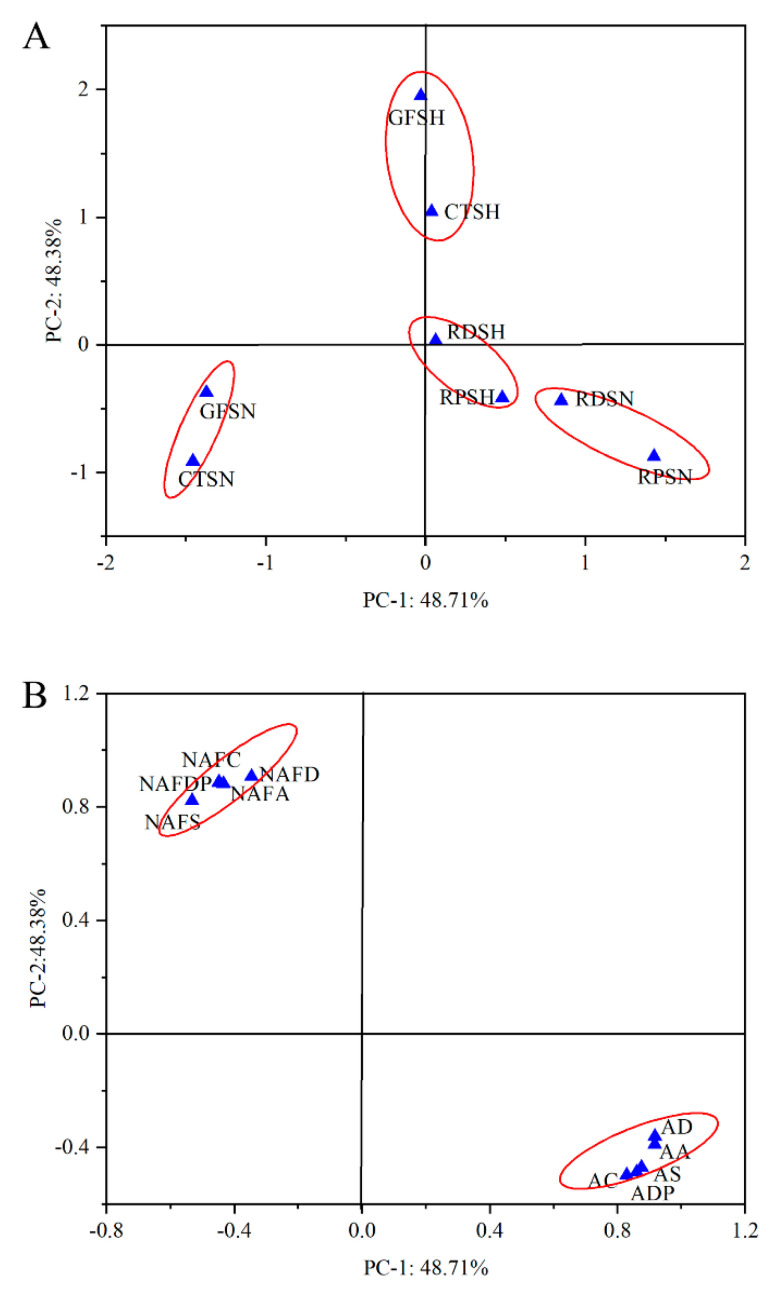
Principal component analysis (PCA) of different blackberry growth stages. (**A**) PCA scores map of different blackberry growth stages. (**B**) Loading map of different blackberry growth stages. Note: NAFD, non-anthocyanidin flavonoid hydroxyl radical-scavenging activities; NAFS, non-anthocyanidin flavonoid superoxide anion radical-scavenging activities; NAFA, non-anthocyanidin flavonoid ABTS radical-scavenging activities; NAFDP, non-anthocyanidin flavonoid DPPH radical-scavenging activities; AA, anthocyanidin ABTS radical-scavenging activities; AS, anthocyanidin superoxide anion radical-scavenging activities; AD, anthocyanidin hydroxyl radical-scavenging activities; ADP, anthocyanidin DPPH radical-scavenging activities.

**Table 1 foods-11-02902-t001:** Variation of non-anthocyanin flavonoid content in Ningzhi 1 and Hull blackberry fruit at different developmental stages.

Compounds	Ningzhi 1				Hull			
	GFSN (μg/g)	CTSN (μg/g)	RDSN (μg/g)	RPSN (μg/g)	GFSH (μg/g)	CTSH (μg/g)	RDSH (μg/g)	RPSH (μg/g)
Kaempferol-3*-O-*rut	0.40 ± 0.01 ^c^	1.04 ± 0.34 ^a^	0.81 ± 0.15 ^b^	0.67 ± 0.03 ^b,c^	38.73 ± 1.78 ^a^	23.00 ± 1.47 ^b^	20.00 ± 1.21 ^c^	3.58 ± 0.43 ^d^
Naringenin	0.12 ± 0.01 ^b^	0.24 ± 0.04 ^a^	0.13 ± 0.02 ^b^	0.11 ± 0.01 ^b^	0.01 ± 0.00 ^b^	0.01 ± 0.00 ^b^	0.02 ± 0.00 ^b^	0.05 ± 0.00 ^a^
Naringenin-7*-O-*glu	2.93 ± 0.81 ^b^	14.73 ± 1.09 ^a^	2.90 ± 0.07 ^b^	2.17 ± 0.20 ^b^	0.79 ± 0.04 ^a^	0.72 ± 0.08 ^a^	0.65 ± 0.00 ^a^	0.76 ± 0.19 ^a^
Quercetin-3*-O-*glu	11.97 ± 0.81 ^b^	14.73 ± 1.09 ^a^	9.30 ± 0.66 ^c^	6.26 ± 0.28 ^d^	173.67 ± 14.98 ^a^	107.63 ± 18.75 ^b^	92.50 ± 8.87 ^b^	43.53 ± 5.46 ^c^
Rutin	0.79 ± 0.03 ^b^	2.40 ± 0.32 ^d^	1.40 ± 0.15 ^c^	3.42 ± 0.21 ^a^	219.67 ± 14.57 ^a^	144.33 ± 11.85 ^b^	119.83 ± 11.79 ^c^	24.47 ± 1.58 ^d^
Procyanidin B2	85.97 ± 3.50 ^a^	52.53 ± 3.35 ^b^	13.63 ± 0.72 ^c^	13.07 ± 1.02 ^c^	147.00 ± 13.23 ^a^	97.33 ± 5.41 ^b^	87.50 ± 4.99 ^b^	45.77 ± 4.21 ^c^
Procyanidin B3	375.33 ± 28.88 ^a^	243.67 ± 14.29 ^b^	76.2 3± 5.23 ^c^	52.17 ± 7.17 ^c^	301.00 ± 15.87 ^a^	218.33 ± 12.22 ^b^	132 ± 16.52 ^c^	65.10 ± 4.06 ^d^
Procyanidin C1	14.87 ± 0.46 ^a^	9.63 ± 0.33 ^b^	1.76 ± 0.13 ^c^	1.60 ± 0.11 ^c^	18.87 ± 0.98 ^a^	10.68 ± 1.01 ^b^	8.50 ± 0.32 ^c^	2.25 ± 0.09 ^d^
NAFC	492.38 ± 25.25 ^a^	327.54 ± 14.73 ^b^	106.17 ± 4.30 ^c^	79.43 ± 6.35 ^d^	899.74 ± 28.75 ^a^	602.04 ± 31.02 ^b^	461.00 ± 2.93 ^c^	185.52 ± 6.15 ^d^

Note: rut = rutinoside; glu = glucoside; For each cultivar, different letters in the same line indicate statistically significant difference at *p* < 0.05 between different developmental stages. Each value represents the mean of three replicates ± standard deviation.

**Table 2 foods-11-02902-t002:** Variation of anthocyanin content in Ningzhi 1 and Hull blackberry fruit at different developmental stages.

Compounds	Ningzhi 1				Hull			
	GFSN (μg/g)	CTSN (μg/g)	RDSN (μg/g)	RPSN (μg/g)	GFSH (μg/g)	CTSH (μg/g)	RDSH (μg/g)	RPSH (μg/g)
Cyd-3*-O-*glu	5.74 ± 0.41 ^c^	14.53 ± 0.50 ^c^	332.00 ± 20.95 ^b^	404.33 ± 12.42 ^a^	5.52 ± 0.21 ^d^	56.60 ± 3.38 ^c^	145.33 ± 13.20 ^b^	507.67 ± 28.53 ^a^
Cyd-3*-O-*sop	0.16 ± 0.01 ^c^	3.06 ± 0.19 ^c^	402.33 ± 9.50 ^b^	423.00 ± 8.19 ^a^	0.08 ± 0.01 ^b^	0.04 ± 0.00 ^b^	0.05 ± 0.00 ^b^	0.65 ± 0.05 ^a^
Cyd-3*-O-*xyl	0.01 ± 0.00 ^c^	0.02 ± 0.00 ^c^	0.44 ± 0.04 ^b^	1.13 ± 0.14 ^a^	0.49 ± 0.01 ^c^	5.07 ± 0.37 ^c^	15.54 ± 0.56 ^b^	193.00 ± 6.24 ^a^
Pelargonidin-3*-O-*glu	0.07 ± 0.00 ^c^	0.24 ± 0.01 ^c^	2.37 ± 0.28 ^b^	7.92 ± 0.32 ^a^	0.01 ± 0.00 ^b^	0.05 ± 0.00 ^b^	0.12 ± 0.01 ^b^	11.28 ± 0.63 ^a^
Petunidin-3*-O-*glu	0.08 ± 0.00 ^c^	0.09 ± 0.00 ^a^	0.10 ± 0.00 ^a^	0.09 ± 0.00 ^b^	0.08 ± 0.00 ^c^	0.08 ± 0.00 ^c^	0.09 ± 0.00 ^b^	0.10 ± 0.00 ^a^
Cyd-3*-O-*rut	0.23 ± 0.02 ^c^	1.19 ± 0.10 ^c^	131.00 ± 5.29 ^b^	186.33 ± 9.07 ^a^	0.36 ± 0.01 ^d^	1.24 ± 0.10 ^c^	1.84 ± 0.19 ^b^	4.50 ± 0.58 ^a^
Ped-3*-O-*gal	0.32 ± 0.01 ^c^	0.34 ± 0.02 ^c^	0.41 ± 0.01 ^b^	0.72 ± 0.05 ^a^	0.42 ± 0.02 ^c^	0.54 ± 0.01 ^c^	0.9 ± 0.02 ^b^	4.92 ± 0.17 ^a^
Ped-3*-O-*glu	0.01 ± 0.00 ^c^	0.03 ± 0.00 ^c^	2.00 ± 0.08 ^b^	3.55 ± 0.04 ^a^	0.06 ± 0.00 ^c^	0.17 ± 0.01 ^c^	0.56 ± 0.04 ^b^	5.33 ± 0.26 ^a^
AC	6.61 ± 0.41 ^c^	19.50 ± 0.54 ^c^	870.66 ± 14.39 ^b^	1027.08 ± 22.40 ^a^	7.03 ± 0.21 ^d^	63.78 ± 3.66 ^c^	164.43 ± 13.85 ^b^	727.45 ± 26.80 ^a^

Note: cyd = cyanidin; glu = glucoside; sop = sophoroside; xyl = xyloside; ped = peonidin; rut = rutinoside; gal = galactoside; For each cultivar, different letters in the same line indicate statistically significant difference at *p* < 0.05 between different developmental stages. Each value represents the mean of six replicates ± standard deviation.

## Data Availability

The data and materials supporting the conclusions of this study are included within the article.

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
