# Peer review of "Composition and Antioxidant Activity of Anthocyanins and Non-Anthocyanin Flavonoids in Blackberry from Different Growth Stages"

_foods, 2022, doi:10.3390/foods11182902_

Round 1

Reviewer 1 Report

This manuscript aims to provide information regarding the relationship between these non-anthocyanins flavonoids, anthocyanins and changes in antioxidant indexes so it could provide a theoretical basis for blackberry fruit harvest.

Please take into consideration the following remarks:

The English language used in this manuscript needs improving. Please do so.

Abstract:

Lines 17-18: You can write the values provided in brackets.

Please insert one sentence regarding the conclusions of your study at the end of this section.

Introduction:

Please provide the exact scientific name of the plant that you studied and provide the genus and family names by using the italic form.

Please provide more information regarding the phytochemical evaluation, ripening stages and harvesting habits of blackberries.

Please cite all the articles that you used, as instructed by the journal’s Instructions for authors section (this suggestion stands for the whole manuscript).

Materials and methods:

Please insert a Reagents section.

Figure 1.- Please provide the measuring unit for the OX axis and remove the red underlining of “Ningzi”.

Please update all the methods provided so that they could be reproduced by other scientists, be more precise (i.e. provide the concentration and quantities of all the reagents used, Lines 100, 102:  12 or 16 hours?, Line 103: how much solution was used for the dilution?, etc.).

Tables 1 and 2 – Please insert in the notes the number of replicas performed and how you expressed the results (usually it is Mean ± SD).

Results and discussion:

Please discus your results by providing links between your findings.

Conclusions:

Please provide future perspectives also.

References:

The references must be written exactly as required by the Instructions for authors section of the journal.

Author Response

Dear Editor and Reviewer: Please see the attachment.

Reviewer 2 Report

The authors provided results on the impact of maturation on the contents of non-anthocyanin flavonoids and anthocyanins in two different cultivars with distinct accumulation profiles in relation to their in vitro antioxidants capacities evaluated by 4 different assays.

The results are interesting for the audience of Foods and provide some elements to propose best prunning period for these two blackberry cultivars.

The presentation of the results is of good quality.

1) I only regret that the "total extract" (ie, before separation of non-anthocyanin flavonoids and anthocyanins) was not assayed for the antioxidants activity. This could provide important comparative basis to evaluate the respective contribution of both fractions toward the antioxidant capacity. It would be good to include these data.

2) "non-anthocyanin flavonoids fraction" and "anthocyanins fraction" should be used instead of "non-anthocyanin flavonoids" and "anthocyanins" in particular in the section headings, since these two fractions are enriched but certainly contain other types of (minor) compounds.

3) References format is not correct.

Author Response

(The authors gave the same response as above.)

Round 2

Reviewer 1 Report

The manuscript was improved.